# Genomic analysis of shiga toxin-containing *Escherichia coli* O157:H7 isolated from Argentinean cattle

**Ariel Amadio**[1⦿], **James L. Bono**[2⦿], **Matías Irazoqui**[1], **Mariano Larzábal**[3], **Wanderson Marques da Silva**[3], **María Florencia Eberhardt**[1], **Nahuel A. Riviere**[3], **David Gally**[4], **Shannon D. Manning**[5], **Angel Cataldi**[3]*

**1** Instituto de Investigación de la Cadena Láctea IDICaL (INTA-CONICET), Rafaela, Argentina, **2** U.S Meat Animal Research Center, Agricultural Research Service, U.S. Department of Agriculture, Clay Center, Nebraska, United States of America, **3** Instituto de Agrobiotecnología y Biología Molecular (IABIMO)-CICVyA, Instituto Nacional de Tecnología Agropecuaria (INTA), Consejo Nacional de investigaciones Científicas y Tecnológicas (CONICET), Hurlingham, Argentina, **4** Division of Immunity and Infection, The Roslin Institute and R(D)SVS, The University of Edinburgh, Easter Bush, Midlothian, United Kingdom, **5** Department of Microbiology and Molecular Genetics, Michigan State University, East Lansing, Michigan, United States of America

⦿ These authors contributed equally to this work.
* cataldi.angeladrian@inta.gob.ar

## Abstract

Cattle are the main reservoir of Enterohemorrhagic *Escherichia coli* (EHEC), with O157:H7 the distinctive serotype. EHEC is the main causative agent of a severe systemic disease, Hemo-lytic Uremic Syndrome (HUS). Argentina has the highest pediatric HUS incidence worldwide with 12–14 cases per 100,000 children. Herein, we assessed the genomes of EHEC O157:H7 isolates recovered from cattle in the humid Pampas of Argentina. According to phylogenetic studies, EHEC O157 can be divided into clades. Clade 8 strains that were classified as hyper-virulent. Most of the strains of this clade have a Shiga toxin stx2a-stx2c genotype. To better understand the molecular bases related to virulence, pathogenicity and evolution of EHEC O157:H7, we performed a comparative genomic analysis of these isolates through whole genome sequencing. The isolates classified as clade 8 (four strains) and clade 6 (four strains) contained 13 to 16 lambdoid prophages per genome, and the observed variability of prophages was analysed. An inter strain comparison show that while some prophages are highly related and can be grouped into families, other are unique. Prophages encoding for stx2a were highly diverse, while those encoding for stx2c were conserved. A cluster of genes exclusively found in clade 8 contained 13 genes that mostly encoded for DNA binding proteins. In the studied strains, polymorphisms in Q antiterminator, the *Q-stx2A* intergenic region and the O and P γ alleles of prophage replication proteins are associated with different levels of Stx2a production. As expected, all strains had the pO157 plasmid that was highly conserved, although one strain displayed a transposon interruption in the protease EspP gene. This genomic analysis may contribute to the understanding of the genetic basis of the hypervirulence of EHEC O157:H7 strains circulating in Argentine cattle. This work aligns with other studies of O157 strain varia-tion in other populations that shows key differences in Stx2a-encoding prophages.

**Data Availability Statement:** All sequence data was deposited in GenBank. All raw sequencing experiments were deposited in SRA. Scripts used are available at https://github.com/arielamadio/

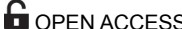

Ecoli_parsing_data Strain BioSample Accession SRA Accession Balcarce_14.2 SAMN17295281 CP076243-CP076244 SRR14419381, SRR14419382 Vac07.1 SAMN17295280 CP076241-CP076242 SRR14419379, SRR14419380 146N4 SAMN17295279 CP076237-CP076240 SRR14419377, SRR14419378 9.1_Anguil SAMN17295278 CP076235-CP076236 SRR14419375, SRR14419376 Balcarce_24.2 SAMN17295277 CP076245-CP076247 SRR14419387, SRR14419388 7.1_Anguil SAMN03470769 CP076232-CP076234 SRR14419385, SRR14419386 Rafaela_II SAMN03470766 CP076230-CP076231 SRR14419383, SRR14419384 438/99 SAMN17295282 JAHCTZ000000000 SRR14419389,SRR14419390.

**Funding:** Fondo para la Investigación Científica y Tecnológica Award Number: PICT 2016 0795 | Recipient: Angel Cataldi https://www.argentina.gob.ar/ciencia/agencia/fondo-para-la-investigacion-cientifica-y-tecnologica-foncyt Fondo para la Investigación Científica y Tecnológica Award Number: PICT-2015-1845 | Recipient: Ariel Amadio https://www.argentina.gob.ar/ciencia/agencia/fondo-para-la-investigacion-cientifica-y-tecnologica-foncyt Agricultural Research Service Award Number: Project Number 3040-42000-017-00-D | Recipient: James L Bono https://www.ars.usda.gov/ The funders had no role in study design, data collection and analysis, decision to publish, or preparation of the manuscript.

**Competing interests:** The authors have declared that no competing interests exist.

## Introduction

Enterohemorrhagic *Escherichia coli* (EHEC), with cattle as the primary animal reservoir, is a globally important foodborne pathogen capable of causing hemorrhagic colitis and hemolytic uremic syndrome (HUS) in humans. EHEC infection in humans can occur via ingestion of food products of bovine origin such as meat, dairy, or by contact with fecal contaminated fruits, vegetables and water sources [1]. Due to its low infectious dose (<100 CFU) and severe clinical outcomes, EHEC infections are considered a serious public health concern. While numerous EHEC serotypes are linked to human infections, strains of serotype O157:H7 cause more severe clinical symptoms, and even HUS, than other EHEC serotypes [2, 3]. HUS caused by EHEC O157:H7 has been reported worldwide [4], with the highest incidence in Argentina [5]. The assessment of molecular characteristics governing pathogenicity and virulence of Argentine EHEC isolates is therefore essential.

The key virulence determinant in EHEC O157:H7 is Shiga toxin (Stx) production, which directly contributes to HUS. Stx genes are encoded on lambdoid phages and EHEC strains can contain one or two Stx subtypes including type 1 (Stx1) or type 2 (Stx2) [6]. Multiple *stx* variants, *stx1* ($stx1_a$, $stx1_c$, and $stx1_d$) and $stx_2$ ($stx2_a$, $stx2_b$, $stx2_c$, $stx2_d$, $stx2_e$, $stx2_f$, $stx2_g$ [7] $stx2_h$ [8] $stx2_i$ [9]) and stx2k [10]) have been reported. According to epidemiological studies, Stx2-producing strains and especially Stx2a, are associated with more severe cases of infection than Stx1-producing strains [11–13].

Another central virulence attribute of EHEC O157:H7 is the pathogenicity island encoding the locus of enterocyte effacement (LEE) [14, 15], which contains genes for a type 3 secretion system (T3SS) and intimin (*eae*), a gene critical for the histological attaching and effacing (A/E) lesions characteristic of EHEC O157:H7. In addition, other important genes involved in EHEC O157 virulence are within the pO157 plasmid [16]. Other factors not yet characterized, however, may also be essential for the full virulence of EHEC O157:H7.

According to phylogenetic studies, EHEC O157 can be divided into three main lineages and nine clades [17, 18]. Various clinical cases on multiple continents and countries have been associated with the clade 8 EHEC strains (I/II lineage). For instance, in Argentina a high prevalence of clade 8 strains was evident in cattle and humans [19, 20]. In the USA, clade 8 strains have been associated with more severe human disease, which led researchers to define it as hypervirulent [18]. Although clade 8 strains had higher Stx2a expression levels relative to strains from other clades, along with unique genetic features [11, 21, 22], to date the factors associated with the increased virulence and Stx2a production are not completely understood. Interestingly, the clade 8 EHEC O157:H7 isolates obtained from bovines in Argentina had a certain degree of genetic variability and displayed variable virulence in *in vitro* and *in vivo* assays [20].

To better understand the molecular bases related to virulence, pathogenicity and evolution of Argentinian EHEC O157:H7, we performed a comparative genomic analysis of these isolates through whole genome sequencing. Notably, we identified conserved regions, insertions/deletions (in/dels) and inversions, as well as variations within core and accessory genes. We also report on the distribution of key virulence genes and the presence of lambdoid phages and plasmids, further highlighting the diversity of the analyzed EHEC O157:H7 isolates.

## Material and methods

### Strains

Eight EHEC O157:H7 isolates previously recovered from cattle of the central Humid Pampas of Argentina between 2002 to 2011, were examined in this study (**Table 1**). Isolates were

**Table 1. Strains with clade assigned and *stx* genotypes.**

| Strain | Clade | *stx* subtypes |
|---|---|---|
| RafaelaII | 8 | stx2a, stx2c |
| Vac07.1 | 8 | stx2a, stx2c |
| Balcarce24.2 | 6 | stx2a, stx2c |
| 9.1Anguil | 8 | stx2c |
| 7.1Anguil | 6 | stx2a, stx2c |
| Balcarce14.2 | 8 | stx2a, stx2c |
| 438–99 | 6 | stx2c |
| 146N4 | 6 | stx1, stx2c |

previously classified into clades by evaluating a subset of 23 single nucleotide polymorphisms (SNPs) [20]from the original 96 SNPs shown to differentiate each clade [18] following annotation (**Table** 1). In addition, six reference EHEC O157:H7 genomes were included in the analysis: EDL933 accession number CP008957 [23], Sakai (NC_002695, [24]), TW14359 (NC_013008, [25]), TW14588 (NZ_CM000662, [26]), 644-PT8 (NZ_CP015831, [27]), SS52 (NZ_CP010304, [28]). The bacteria were grown at 37˚C on Luria-Bertani (LB, Difco Laboratories, USA) agar plates or aerobically in LB broth.

## DNA preparation and PacBio whole-genome sequencing

An aliquot of 250 μl of an overnight culture of the different EHEC O157:H7 strains was added to 20 mL of LB and incubated at 37˚C with shaking for 3 ½ h. The bacteria were then harvested for DNA extraction using Genomic-Tip 100/g (Qiagen Inc Valencia, CA. DNA (10 ug/ml) were sheared to a targeted size of 30 kb using a g-TUBE (Corvaris, Woburn, MA) and subsequently concentrated using 0.45X volume of AMPure PB magnetic beads (Pacific Biosciences, Menlo Park, CA) according to the manufacturer's protocol.

Sequencing libraries were created using 5 μg of sheared, concentrated DNA and the PacBio SMRTbell Template Prep Kit 1.0, according to the manufacturer's protocol. Each library was sequenced using the RS II sequencing platform (Pacific Biosciences, PacBio) with the P6/C4 sequencing chemistry and the 360 min data collection protocol.

## PacBio sequence assembly into closed circularized genomes and annotation

PacBio reads were assembled using HGAP3 (SMRTanalysis Version 3.0, Pacific Biosciences) and the resulting contigs were imported into Geneious software (Biomatters, Ltd., Auckland, New Zealand). If present, overlapping sequences on the ends of the contigs were removed from the 5' and 3' ends to generate circularized chromosomes and plasmids using the software Geneious (Biomatters, Ltd.). The closed chromosomes were reoriented to start with the putative origin of replication using Ori-Finder 2 [29]. The closed chromosomes and plasmids were polished twice for accuracy using the RS_Resequencing 1.0 protocol in SMRTanalysis and by mapping error corrected PacBio reads to the chromosomes and plasmids using Geneious software (Biomatters, Ltd.). Genome sequences are deposited in GeneBank (BioProject PRJNA280853). Automatic annotations were performed using Prokka [30] against a reference set of six O157:H7 genomes as primary annotations described in (see the Strain section in M&M). Protein sequences were extracted from those genomes and clustered at 95% identity using cd-hit [31] to create a trusted source for annotation for Prokka.

## Whole genome comparison

Genomes were compared using BLAST+ [32] and visualized in Artemis Comparison Tool software [33]. Comparison files were generated online at WebACT [34] or locally.

## Prophage prediction and extraction of virulence genes

Phaster webserver was used to predict putative prophage sequences in the bacterial genomes [35]. Subsequently, prophage sequences, including Stx prophages, were extracted from the bacterial genome sequences using Artemis [36]. Only prophages identified as complete and questionable by Phaster software were considered for the analysis. Prophages were named R1 to Rn as they appear from 5´to 3´ in the linear representation of the chromosome (with origin of replication at the 5' end). For example, RafaelaII-R2 means it is the second phage identified in strain RafaelaII.

Prophage comparisons were carried out using Blast+ [32] as follows: each prophage sequence was extracted and used as a query for comparison with all prophages from the same strain (Intra-strain) or from the other strains (Inter-strain). The output was processed with custom Perl scripts (https://github.com/arielamadio/Ecoli_parsing_data) to obtain an average of identity for the High Scoring Pairs (HSPs) and the coverage for each prophage as the 'qcovus' parameter (query coverage per unique subject). To have an overall estimation of phage similarity across strains, prophage comparisons were performed with settings Query Coverage Per Subject 80% and not restricting the other parameters (query Coverage Per Unique Subject, Average Identity and HSPs number). These criteria were applied in inter-strain prophage diversity section. To classify the prophages into families by similarity, comparisons were performed with hits Query Coverage Per Subject 80% and query Coverage Per Unique Subject 90% and not restricting the other parameters (Average Identity and HSPs number). Also, a maximum tolerance of 8kb between query and subject sizes was selected.

An even stricter relatedness criterion (query coverage 80% and Query coverage Per Unique Subject 90% and gaps and inversions < 4kb) was established to identify distributed identical prophages. For the nomenclature of these identical prophages, the name of the strains with the identical prophage is followed by the letter and number of the family at the end of the name. For example, TW14359-R3 denotes the third phage in TW14359 and belongs to family R and would be considered identical to Vac07.1-R3, RafaelaII-R3 and Balcarce14.2-R3 (**S1 Table**).

Virulence gene sequences were identified using Virulence Finder 2.0 [37], with a nucleotide identity threshold of 90% and a minimum length of 60%. The virulence genes considered in this analysis were those encoded on the LEE (*tir*, *eae*, *espA*, *espB* and *espF*) or those that coded for toxins (*stx2* and *stx1*), effectors (*tccP*, *espJ*, *nleA*, *nleB1*, *nleB2* and *nleC*), stress resistance proteins (*gadA* and *gadB*) and the adhesin *iha* [37].

## Prediction of clusters of orthologous genes and pan-genomic analysis

Roary [38] was used to identify the pan-genome from the genomes of strains sequenced in this work and the set of references used for annotation. All genomes were re-annotated with Prokka (with a reference set of proteins) previous to running the pangenome pipeline to standardize the annotations.

Comparisons of gene content between clades were performed using "difference" method in Roary and by specifying the groups to compare. A tree was constructed with the alignment of the core genome using RAxML [39] with GTRGAMMA as model and 1000 bootstraps. Clade assignment by SNP analysis were determined with NucDiff [40] by comparing the complete genomes with the strain Sakai as a reference. Then, the 23 SNPs corresponding to the

genotyping panel [18] were extracted from the comparison output to subsequently infer the strain genotypes and clades.

### Shiga-toxin detection and cytotoxicity in Vero cells

Stx expression was determined for filtered culture supernatants with or without mitomycin C induction, by using RIDASCREEN Kit Verotoxin enzyme immunoassay (R-Biopharm Latin America) and then semi-quantified, as previously described [20]. To measure Vero cell cytotoxicity, the strains were cultured overnight at 37°C in 5 ml of LB broth and the culture was centrifuged at 200 rpm. The supernatant was filtered (0.22 μm filters) and subsequently assayed for cytotoxicity on Vero cells, as previously described [20].

## Results and discussion

### General genomic features of the EHEC O157:H7 isolates

Complete chromosomes and plasmids were obtained for all but one of the eight analysed strains (**Table** 2). The chromosome of Strain 438–99 assembled as two contigs that could not be resolved further. Chromosome sizes and gene content of the sequenced strains ranged from 5415530 bp to 5750490 bp and 5193 ORFs to 5599 ORFs, for strains 7.1Anguil and 9.1Anguil, respectively. All the strains characterized here and the reference strain TW14359 belong to lineage I/II, except for the reference strain EDL933, which is from lineage I [41, 42]. Whole genome multiple alignments showed a high level of conservation around the origin of replication and variation around the replication terminus. Most of the In/Dels and inversions were associated with the presence of lambdoid prophages (**Fig 1**), and located mostly around the

**Table 2. Chromosome and plasmid characteristics from the sequenced strains and two previously published strains.**

| | Tw14359 (cl8)[a] | RafaelaII (cl8) | Vac07.1 (cl8) | Balcarce24.2 (cl8) | 9.1Anguil (cl8) | Balcarce14.2 (cl6) | 7.1Anguil (cl6) | 146N4 (cl6) | 438–99 (cl6) | EDL933 (cl3) |
|---|---|---|---|---|---|---|---|---|---|---|
| Chromosome size (bp) | 5528136 | 5544602 | 5425421 | 5457079 | 5415130 | 5666506 | 5750490 | 5528657 | 5653150 | 5547323 |
| Number of Genes | 5253 | 5348 | 5219 | 5224 | 5193 | 5479 | 5599 | 5318 | 5465 | 5675 |
| Coding bases | 4728279 | 4857984 | 4752591 | 4781259 | 4745568 | 4952364 | 5011443 | 4844244 | 4946871 | 4819848 |
| genes per kb | 0.95 | 0.964 | 0.961 | 0.957 | 0.958 | 0.966 | 0.973 | 0.961 | 0.966 | 1.023 |
| (bases per gene): | (1052) | (1036) | (1039) | (1044) | (1042) | (1034) | (1027) | (1039) | (1034) | (977) |
| Gene average length | 900 | 908 | 910 | 915 | 913 | 903 | 895 | 910 | 905 | 849 |
| coding percentage | 85,5 | 87,6 | 87,5 | 87,6 | 87,6 | 87,3 | 87,1 | 87,6 | 87,5 | 86,8 |
| GC percentage | 51,63 | 51,57 | 51,56 | 51,67 | 51,6 | 51,51 | 51,53 | 51,64 | 51,48 | 51,56 |
| rRNA bases (number) | 32078 (22) | 31966 (22) | 31964 (22) | 31964 (22) | 31964 (22) | 36515 (25) | 31963 (22) | 31962 (22) | 31964 (22) | 32223 (22) |
| tRNA bases (number) | 8246 (106) | 8272 (106) | 8042 (103) | 8042 (107) | 8195 (105) | 9055 (116) | 8425 (108) | 7965 (102) | 8337 (107) | 7579 (101) |
| lambdoid phages | 15 | 14 | 13 | 14 | 13 | 15 | 15 | 16 | 16 | 14 |
| Plasmids | 94,601bp (pO157) | 94,756bp (pO157) | 94,989bp (pO157) | 95,209bp (pO157) 3,306bp | 94567bp (pO157) | 94,595bp (pO157) | 96510bp (pO157) 55,027bp | 95343bp (pO157) 7,323bp 3,306bp | 91,463bp (pO157) 56,932bp | 92,076bp (pO157) |

[a]cl = clades

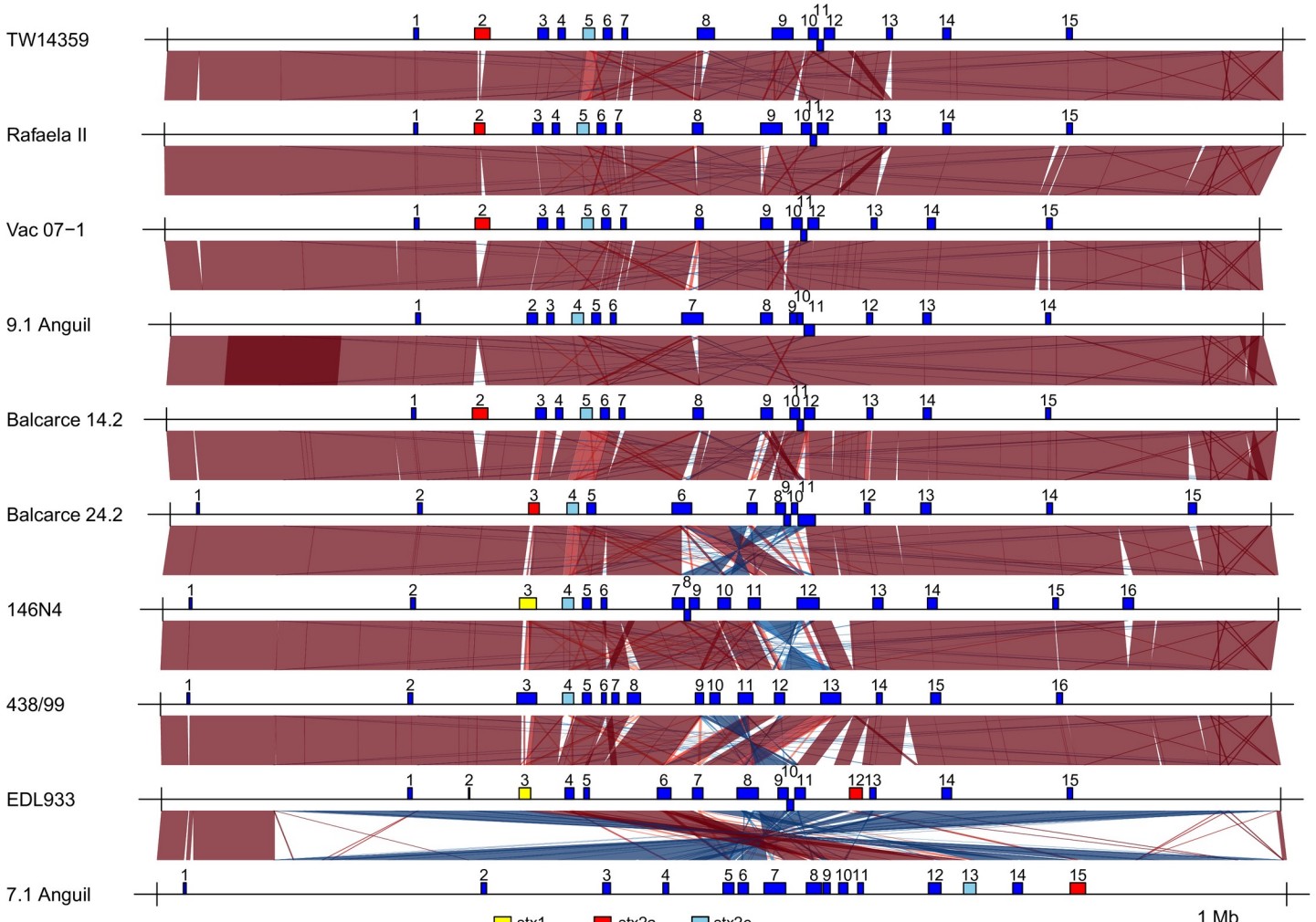

**Fig 1. Schematic lineal representation of aligned EHEC O157 genomes.** The origin of replication is placed at the 5´and 3'end. Lambdoid prophages are represented as boxes of blue color, except for those carrying *stx* genes. In those cases, *stx1*, *stx2$_a$* and *stx2$_c$* prophages were indicated in yellow, red and light blue boxes, respectively. Artemis comparison tool was used to indicate regions of similarity (red), inversions (blue) and In/dels (clear).

terminus of replication of the genome. EHEC O157:H7 7.1Anguil had a large, inverted region comprising 5 Mb, except at the origin of genome replication. This inversion can also be interpreted as involving the origin of replication, but that is against the convention of linear representation of genomes using *ori* as the first nucleotide. Two approaches followed to assess and confirm the presence of this inversion. The first approach consisted of a manual inspection of long reads, whereas the second involved performing long PCR with two pairs of primers located outside of the inversion (and the repeated region) (**S1 Fig**) to amplify a 6kb region on both sides of the inversion point. Importantly, the use of PacBio long read sequencing technology allowed us to reconstruct complete finished circular chromosomes and plasmids for most of the analysed isolates, with the subsequent possibility of including genome features, such as phage content and position, in the analysis.

The strains 146N4 and 438–99 from clade 6 presented several reorganizations in the terminus region of the chromosome relative to other strains. The reorganized regions were contained between prophages R11 and R12 in 146N4 and R9 to R11 in 438–99. The chromosomal

sequence is conserved in the inverted regions among all strains with the difference in the chromosomal architecture. On the other hand, 7.1 Anguil, has an inversion that isn't not associated with a phage region but with a ribosomal RNA region. This large inversion the sub-region between R12-R13 from 438–99 is moved to the 5' of the regions for all strains. This sub-region, however, is inverted between 146N4 and 438–99 and is different from that of other strains. In addition, 438–99, EDL933 and 7.1Anguil shared the orientation of this sub-region, whereas 146N4 shared the direction of the remaining strains (**Fig 1**).

The number of inserted prophages detected by Phaster, varied from 13 to 16 per genome. Lambdoid prophages were located between 1.2 Mb to 4.60 Mb of the genome coordinates. 149 prophages were identified in the strains, including the reference strains (**S1 Table**). The number and sequence variability of lambdoid prophages among our EHEC O157:H7 strains supports previous reports [11, 12, 43–46]. The order of prophages in the genome has some conservation, as they occupy main preferential integration sites. Most of the sequenced strains had two Stx-converting lambdoid prophages, except EHEC O157:H7 438–99 that had one Stx2c-converting lambdoid prophage, and 9.1Anguil (clade 8) that had probably lost the Stx2a prophage. Stx2a encoding prophages were integrated at *argW* tRNA in clade 8 strains and 7.1Anguil, whereas it was integrated at *yehV/mirA* and *wrbA* in Balcarce24.2 and EDL933, respectively. The gene content of prophages encoding for $stx2_a$ are hypervariable. No $stx2_a$ containing prophage exhibited more than 70% conservation among the analysed genomes.

There was no genomic inversion larger than 50 kb among the clade 8 strains, in contrast to those of the strains of other clades. *E. coli* inversions are located around the terminus of replication, a region that seems prone to rearrangements [47] and Fitzgerald, personal communication. The 7.1Anguil strain showed the largest inversion in rRNA genes, which is a well described event in *E. coli* [48]. By contrast, the locations of the inversions for the remaining strains were restricted to lambdoid phages, which have an ad-hoc machinery of recombination that promotes inversions [49].

## Prophage diversity

Inter-strain prophage diversity: some strains contained ubiquitous prophages with high level of similarity in relation to prophages from all strains. Balcarce24.2 and 7.1Anguil had five prophages with similarity hits in all strains whereas some clade 8 strains (Vac07.1, Tw14359, 9.1Anguil) had four conserved prophages (**S2 Fig**).

Intra-strain prophage diversity similarities only occurred with less than 85% coverage and 85% identity, as determined by blastn with all the prophages from a given strain. This is expected due to the immunity system of lambdoid phages [50]. However, a set of prophages had similarities covering between 70% to 75% of their sizes in other prophages in the same strain. For example, in the strains Vac07.1, Tw14359 and Balcarce14.2, prophages R13 and R6 were similar, whereas in 9.1Anguil, prophage R12 was similar to R5. Similarly, intra-strain prophage similarity was observed with R14 and R5 in 438–99; R13 and R4 in EDL933, R13 and R10 in RafaelaII and R12 and R5 in Balcarce24.2. Also, 438–99 showed two adjacent prophages (R6 and R7) with high level of similarity, which did not occur in any other strain. 438–99 was the strain with the highest level of intra strain phage similarity. Strains with less intra strain phage similarity were Balcarce14.2 and RafaelaII strains (**S2 Table**).

In order to identify highly related prophages across the strains, stricter relatedness parameters were used. Including visual inspection for large gaps and inversions in dot plots of paired genomes. The analysis identified 21 families with two to ten prophages in each family that accounted for 120 prophages (**S1 Table**). The remaining 29 prophages were unique to a particular strain. The families with 6 to 10 prophages are described in (**S3 Table**). Beside these families

there are another 10 smaller families composed of 2 to 5 prophages. Interestingly, in some families, prophages have relevant metabolic or regulatory genes that are located 5' or 3' of prophage genes but are located between the predicted *attL* and *attR* integration sites (**S3 Table**).

**Unique prophages.** Some prophages are unique to a single strain (with no other hits above the established cut off) (**S1 Table, S2 Fig**). Strains with more unique prophages are 146N4 (six) and Balc24.2 (four). Clade 8 strains tend to have a lower number of unique prophages, but this may be due to the higher frequency of clade 8 strains in the study.

**Stx2 encoding prophages.** The diversity of Stx2a encoding prophages is reflected by the fact that prophages R2 from TW14359, Vac07.1 and Balcarce14.2 belong to family B, R3 from Balcarce24.2 belongs to family C and R2 from RafaelaII, R12 from EDL933 and R5 from 7.1Anguil are unique. This result strongly differed from that of prophages encoding $stx2_c$ that belongs to family E. In contrast to the $stx2_a$ containing prophage, the prophages containing the $stx2_c$ gene were highly conserved among the studied strains. All nine $stx2_c$ prophages showed an average similarity of 97.7% ±2.13%, whereas the seven $stx2_a$ prophages displayed an average similarity of 60.5% ± 6.3%.

**Identical prophages.** We determined if the same prophage is integrated in the genome of different strains using a stricter relatedness criterion than that for grouping the prophages in families (**S1 Table**), We identified 16 shared types of prophages (identical prophages in sequence and size present in more than one strain).

The most distributed shared types in this set of strains were TW14359F6 (in all ten strains), TW14359E5 (eight strains) as well as TW14359J10 TW14359K11 and TW14359M13 (all present in seven strains). With these prophage sequences, we performed megablast against nr NCBI database with highly restrictive search parameters coverage 98%, identity 99% (**S4 Table**). The shared types most frequently found were Tw14359N14, Tw14359J10; Tw14359A1; Tw14359F6 with 179, 175, 170 and 160 hits, respectively followed by other less represented shared type prophages in the nr database. This method does not allow to determining the size of the targeted prophages in the nr database but indicates that the sequence of the prophages studied here are found as complete prophages in other genomes.

As reported by other authors [11, 12, 43–46], extreme phage diversity was observed probably because lambdoid phages are equipped with specialized recombination machinery as Exo, Bet, Gam and other proteins [50]. However, in spite of this general diversity they could be grouped by sequence alignments in families of high similarity and even in individual identical sequences shared by different genomes. A tendency was observed that shorter prophages are likely to be associated with higher clustering (**S3 Fig**), as previously described by Shaaban et al. [46]. Conversely, Larger prophages (>80 kb) were overrepresented among the unique prophages.

## Pan genomic analysis reveals regions exclusive to clade 8 strains

The genomes of all sequenced strains and the six strains used as references had 4,415 core genes and a pangenome of 7,881 genes. Roary was used to identify a set of genes only present in the clade 8 genomes. Although no long contiguous exclusive fragments common to all clades 8 strains were detected in the analysis, a short contiguous gene cluster of ten genes was exclusive to this clade. In RafaelaII strain (used as a prototype clade 8), this cluster corresponds to genes from RafaelaII_01805 to 17 and showed 100% identity and conserved synteny in the other clade 8 strains studied here. This segment is located in R3 prophage (family C) of RafaelaII strain (**Table 3**).

The pangenome analysis, even using a small number of genomes, was consistent with previous studies. For instance, the 4415 core genes are very similar in number to the 4369 core-genome identified in 185 strains from the UK [51]. A thorough analysis of the hypervirulent clade 8 strains to search for exclusive regions revealed only one short region that encode for

**Table 3. A ten gene region exclusive to clade 8 strains.**

| Gene[a] | Annotation[b] | BlastP descriptions | Location (start codon base)[d] | Orthologue in EHEC O157 TW14359[e] |
|---|---|---|---|---|
| *RafaelaII_01805* | hypothetical protein | ASCH domain containing protein | 1834934 | ECSP_2983 |
| *RafaelaII_01806* | hypothetical protein | N-acetyl transferase | 1835289 | ECSP_2982 |
| *RafaelaII_01807* | helix-turn-helix transcriptional regulator | helix-turn-helix transcriptional regulator | 1836431 | ECSP_2981 |
| *RafaelaII_01808* | Antirepressor | helix-turn-helix transcriptional regulator | 1837225 | ECSP_2980 |
| *RafaelaII_01812* | ren protein | ren protein | 1837225 | ECSP_2973 |
| *RafaelaII_01813* | multidrug efflux protein | SMR multidrug resistance protein | 1840145 | ECSP_2972 |
| *RafaelaII_01814* | DLP12 prophage recombinase | Recombinase family protein | 1840734 | ECSP_2971 |
| *RafaelaII_01815* | kinase inhibitor | Far kinase inhibitor ybcL | 1842725 | ECSP_2970 |
| *RafaelaII_01816* | DNA-binding transcriptional regulator | helix-turn-helix transcriptional regulator | 1843286 | ECSP_2969 |
| *RafaelaII_01817* | hypothetical protein | DNA base flipping protein. NinB superfamily | 1844298 | ECSP_2968 |

[a]Gene name from RafaelaII.

[b]Annotation as obtained by Prokka.

[c]Blastp top hit against Genbank nr database.

[d]Location in RafaelaII chromosome.

[e]Name of orthologue in *E. coli* O157:H7 strain TW14359.

DNA binding proteins, putative RNA binding domains and a ren protein involved in phage exclusion and protection from RecAB recombinase [52]. The product of many of these genes may have a regulatory effect of prophage gene expression as three of the genes are annotated as transcriptional regulators.

**SNPs/Phylogeny.** A tree constructed using all genes corresponding to the core genome of the analysed and reference strains retrieved four major clusters (**Fig 2**). One clustered contained reference strains Sakai, EDL933 and TW14588, from clades 1, 3 and 2, respectively. The second cluster contained clade 6 strains 7.1Anguil, Balcarce24.2 and the non-*stx2$_a$* 438–99 and 146N4. A third cluster consisted of clade 8 isolates Vac07.1 and RafaelaII. A fourth cluster comprised clade 8 strains 9.1Anguil, Balcarce14.2 and reference strains TW14359 and SS52.

## Plasmid pO157

The isolates contained the megaplasmid pO157 that showed a high degree of conservation (**Fig 3**). EDL933 pO157 plasmid was smaller (92 kb), whereas the pO157 from other strains ranged from 94.5 to 96.5 kb. This longer length compared to pO157-EDL933 is due to a transposon and IS elements insertions after *espP* and before the lipid A modification enzyme genes cluster (**Fig 3**). Other genetic differences also related to the insertion of transposons in pO157 include the pO157 from strain 146N4. It had two transposons inserted at position 80800 at the 5´of the *espP* gene that resulted in two EspP proteins: EspP1 comprising the first 287 N-terminal residues and EspP2 composed of the last 1080 C-terminal fragment. These two fragments are probably non-functional as the segmentation interrupted the Peptidase S6 superfamily domain. EspP has been implicated in inhibiting complement activation, which allows the progress of EHEC infection [53, 54]. In EPEC, EspC, the orthologous of EspP, is related to epithelial cell severe alterations [55]. The loss of a functional EspP protein suggests that this strain my not cause disease in humans or may cause a milder form.

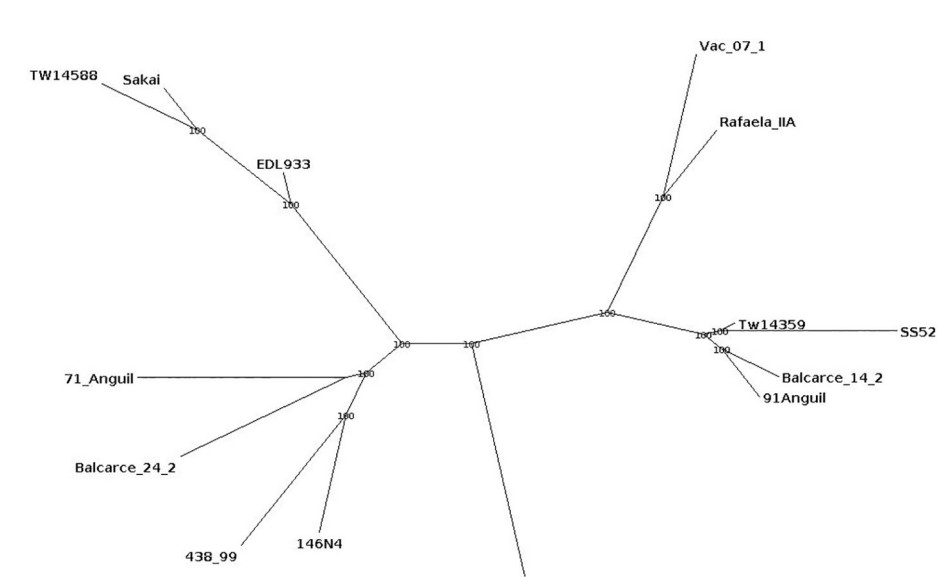

**Fig 2. Phylogenetic tree constructed with the core genome from the Argentinian strains and previously published strains using RAxML.** Bootstrap values are located at each branch point.

## Identification of additional plasmids

Two strains, 7.1Anguil and 438–99, also contained a 55 kb conjugative plasmid, p55 (**Table** 2). Both plasmids shared 99% similarity with 96% coverage according to blastn. These plasmids were similar to pMCR-1 from EHEC O157:H7 strain 2017C-4109 that carries the *mrc-1* gene that mediates colistin resistance [56]. The *mcr1* gene was absent from plasmid p55 from strains 7.1Anguil and 438–99 strains. However, the sequence 5' and 3' to the gene were conserved in p55 and pMCR.1 (**Fig 4**). The annotation of the ORFs of p55 was related mostly to hypothetical proteins and the conjugation machinery. Other plasmids highly similar to p55 from *E. coli* are TP114 from *E. coli* K12 (MF521836); pChi7122-3 from avian enteropathogenic *E. coli* (APEC) 7122 O78:K80:H9 (FR851304); AR_0011 plasmid tig00013784_pilon (CP024858; plasmid pF7386-2 from EHEC O157:H7 (CP038361); an unnamed plasmid from *E. coli* O177:H21 (CP016547) and plasmid pMRSN346355_65.5 (CP018124.).

In addition, strain 146N4 carried a 7 kb plasmid, p7, which contained genes for colicin D synthesis, lysis protein for colicin release and mobilization genes. This plasmid was highly similar to the plasmid pF1273-2 (accession number CP038377). A small 3.3 kb plasmid that was also detected in Balcarce24.2 and 146N4. This plasmid was identical to pOSAK [57–59] and carried the toxin endoribonuclease *lsoA*, an antitoxin *lsoB* genes and the mobilisation gene *mobA2*.

## Inference of distinctive virulence functions based on gene content and polymorphisms

In a previous study, we had described the behaviour of the same set of EHEC O157:H7 strains from cattle in virulence correlated assays, such as Vero cell cytotoxicity, Red Blood Cell (RBS) lysis produced by the T3SS, adherence to different cell lines and lethality, weight loss and tissue damage in inoculated mice [20]. The following section refers to an analysis of some of the genes involved in these pathogenic processes.

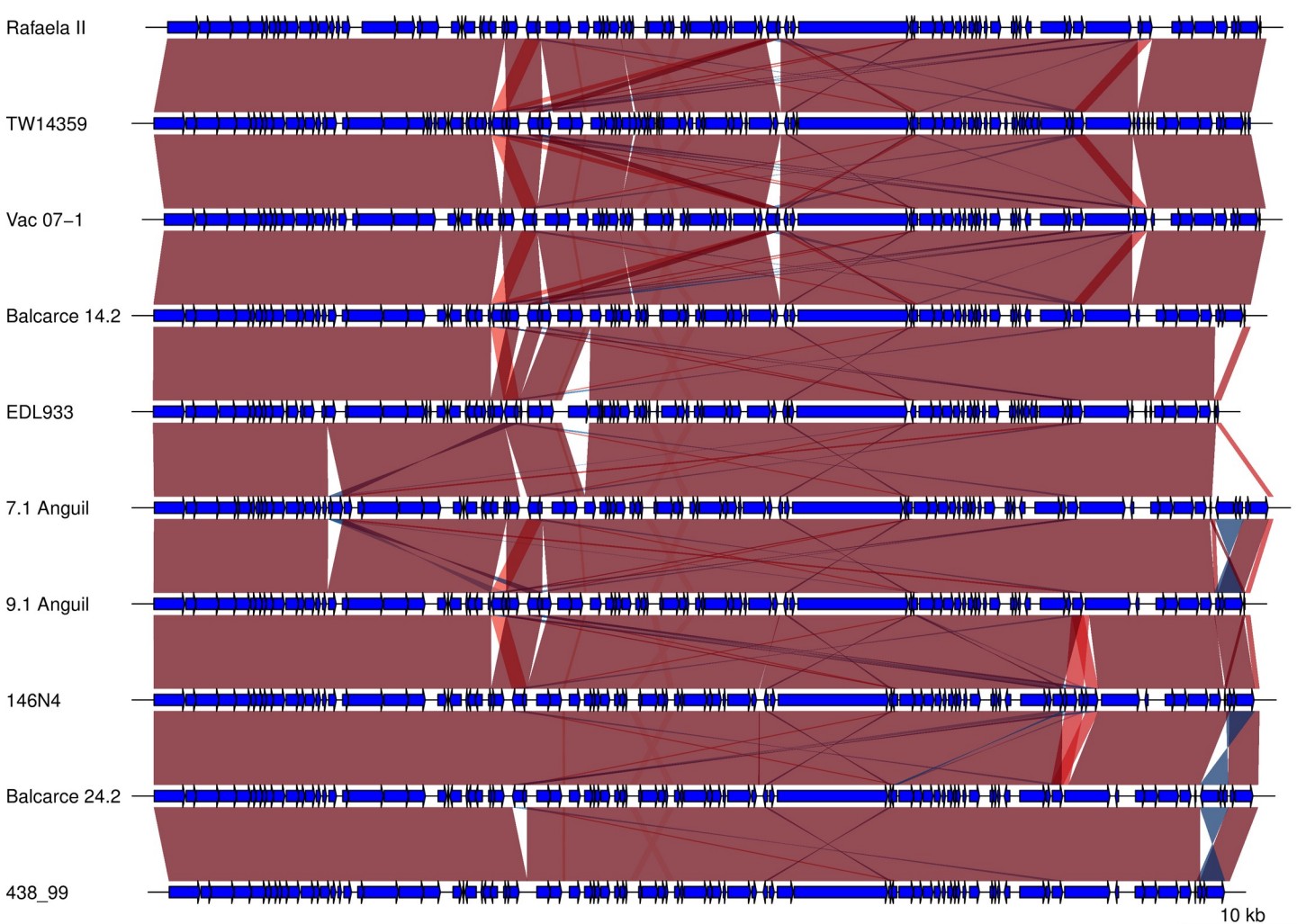

**Fig 3. Alignment of pO157 plasmids from EHEC O157 isolates.** Plasmid are represented in lineal forms with the ORF colored blue. The *tagA* gene was oriented at the 5´end to allow for comparison of plasmid architecture. Artemis comparison tool was used to indicate regions of similarity (red), inversions (blue) and In/dels (clear).

**Shiga toxin encoding prophages and determinants of *stx2* expression.** We comprehensively evaluated the Stx prophages incorporated into each EHEC genome. The strains of clade 8 and 6 had prophages encoding *stx2_a* and are frequently the second lambdoid prophage in the genome (R2, counting from the replication of origin) **Fig 5**. Stx toxin production was analyzed in strains carrying the stx2a operon (**Fig 5**) in order to associate gene content of the encoding prophages with Stx production.

Interestingly, the Stx2a prophage from RafaelaII (**Fig 5**), is a putative defective phage because it lacked the genes for endolysin, terminase, portal protein (a major capsid protein) and the tail fiber protein. All of these proteins are related to the lytic machinery and the bacteriophage body. While one of the isolates, 9.1Anguil (clade 8) had no Stx2a phage. The phage may have been lost before isolation, as suggested by its low Stx2 activity to Vero cells and by the fact that there was a complete *argW* tRNA, the integration site of Stx2a prophage.

Two strains, 9.1Anguil and RafaelaII, are of particular interest because they either lack the *stx2_a* containing prophages or the phage is defective. Sequence analyses indicated that 9.1Anguil does not have a *stx2_a* containing prophage. This is different from most the clade 8

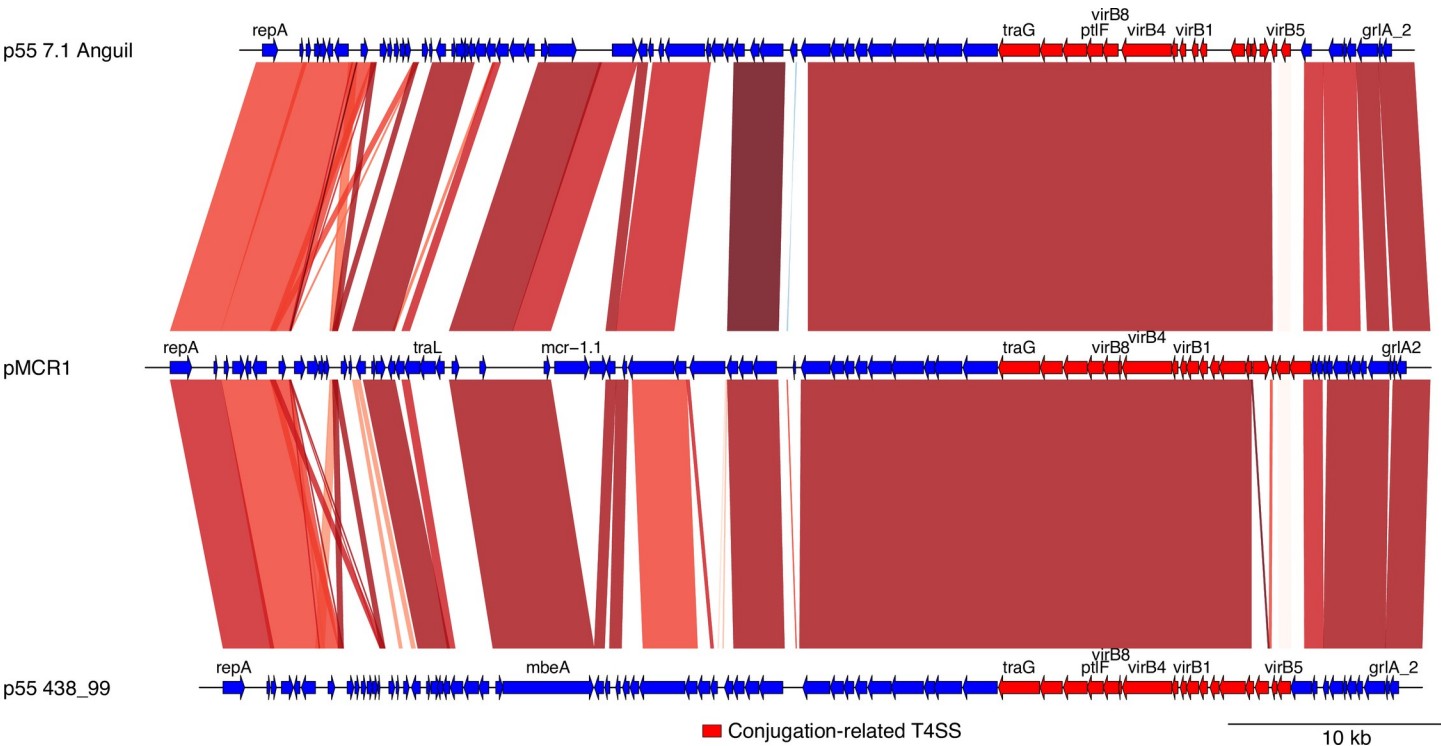

**Fig 4. Alignment of the p55 plasmid of strains 7.1Anguil and 146N4 with a related pMCR1plasmid.** Plasmid are represented in lineal forms with blue and red arrows indicating genes. Artemis comparison tool was used to indicate regions of similarity (red), inversions (blue) and In/dels (clear).

strains as they carry both $stx2_a$ and $stx2_c$ containing prophages [18]. We presume that there was a precise excision of the $stx2_a$ containing prophage in 9.1Anguil that didn't leave any phage sequence in the chromosome when it excised. However, we can't rule out that 9.1Anguil had never had a $stx2_a$ containing prophage which would make it an interesting strain for further study. With only the $stx2_c$ gene, Stx2 production of this strain was extremely low. This finding suggests that almost all the production of Stx2 in $stx2_a$-$stx2_c$ strains is due to the $stx2_a$ containing phage, as previously demonstrated [60], and that the low expression of $stx2_c$ gene is not due to a repression exerted by the $stx2_a$ prophage. In turn, RafaelaII ($stx2_a$-$stx2_c$) contains a defective $stx2_a$ containing prophage that lacks the genes encoding for essential components of the phage body and lytic machinery. However, this strain produces high levels of Stx in culture supernatants, as shown by ELISA and verocytotoxicity assay. This observation suggests a complete prophage is not needed for Stx2 production as long as the regulatory genes are present.

Previously published research demonstrated that different alleles of the $Q$ antiterminator gene affect the expression level of the $stx2a$ operon [61]. Stx2a prophages that carry the $Q_{933}$ allele are related to high expression, while those with $Q_{21}$ [62] and $Q_{111H-}$ [60]. are linked to low expression. All the strains studied here had the $Q_{933}$ allele in the Stx2a prophage as determined by *in silico* PCR with primers described by Olavesen et al. [63]. However, the analysed strains presented an important variation in Stx2 production level. The analysis of the $Q$ gene sequences showed previously undescribed polymorphisms in the $Q_{933}$ allele of $Q$ gene. These $Q_{933}$ polymorphisms are evident in two strains, Balcarce24.2 and 7.1Anguil (**Fig 6A**). Interestingly, 7.1Anguil had the highest cytotoxicity to Vero cell line. Also, strains 7.1Anguil and Balcarce24.2 shared other mutations in the pR'-tRNA region Q-Stx2A intergenic region, in

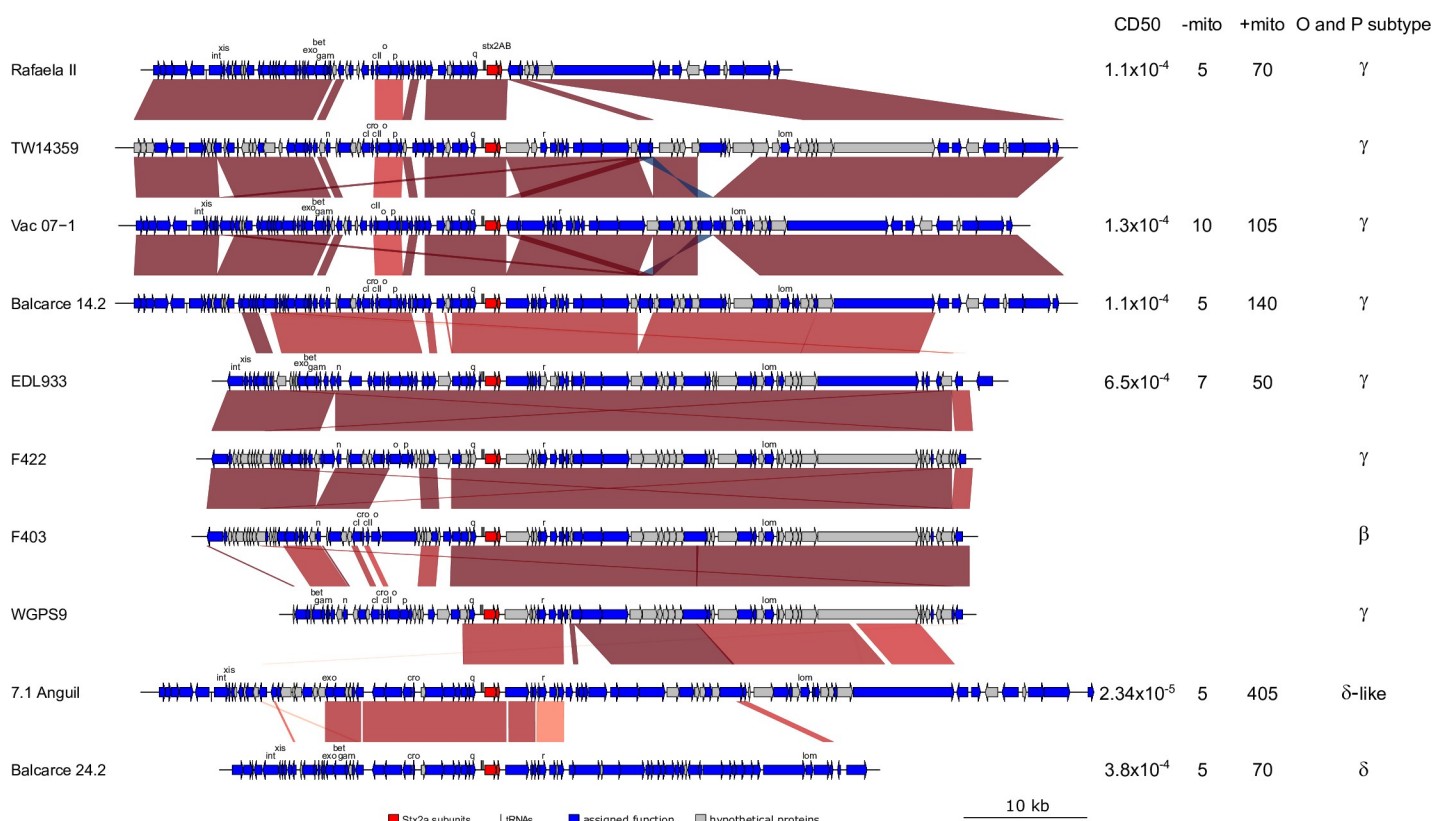

**Fig 5. Alignment of prophages carrying *stx2ₐ*, the red arrows represent genes of the *stx2ₐ* operon; the grey arrows indicate hypothetical proteins, and the blue arrows refer to genes encoding for protein with assigned function.** Some relevant genes of lambda phage are marked. Artemis comparison tool was used to indicate regions of similarity (red), inversions (blue) and In/dels (clear) or no homology (no lines). Columns at the right show cytotoxicity in Vero cells, by ELISA with or without mitomicin and O and P protein subtype.

comparison to the other strains, including the reference strains (**Fig 6B**). The detected non-synonymous mutations in *Q* genes that may change the function of the protein as well as the SNPs in the intergenic region between the *Q* and *stx2* genes deserves more research to assess the effect of these substitutions on *stx* expression.

As replication proteins O and P from lambdoid prophages encoding *stx2* have been implicated in Shiga toxin expression [11], we subsequently assessed these genes, which are located upstream of *stx2ₐ*. Most of the strains had the γ allele of O and P genes, except Balcarce24.2 that carry the δ allele and 7.1Anguil that possesses a novel δ allele variant that we called δ–like allele (**Fig 5**). This variant of O and P allele has not been described previously.

Interestingly, most of the strains of our study that have *stx2ₐ* genes have the subtype γ, except strains Balcarce24.2 (clade 6) and 7.1Anguil (clade 6). The O-protein δ-like subtype from the highest Stx producer 7.1Anguil has seven amino acid substitutions in relation to the canonical δ subtype, whereas the P-protein from δ-like subtype has eight substitutions concentrated in the N-terminal end. The role of these substitutions in O and P function remain to be elucidated. The O- and P-proteins, which have been proposed to have primase and replicase activity, respectively, are the only proteins from a lambdoid phage implicated in replication as the DNA polymerase and other constituents of the replication machinery are provided by the host.

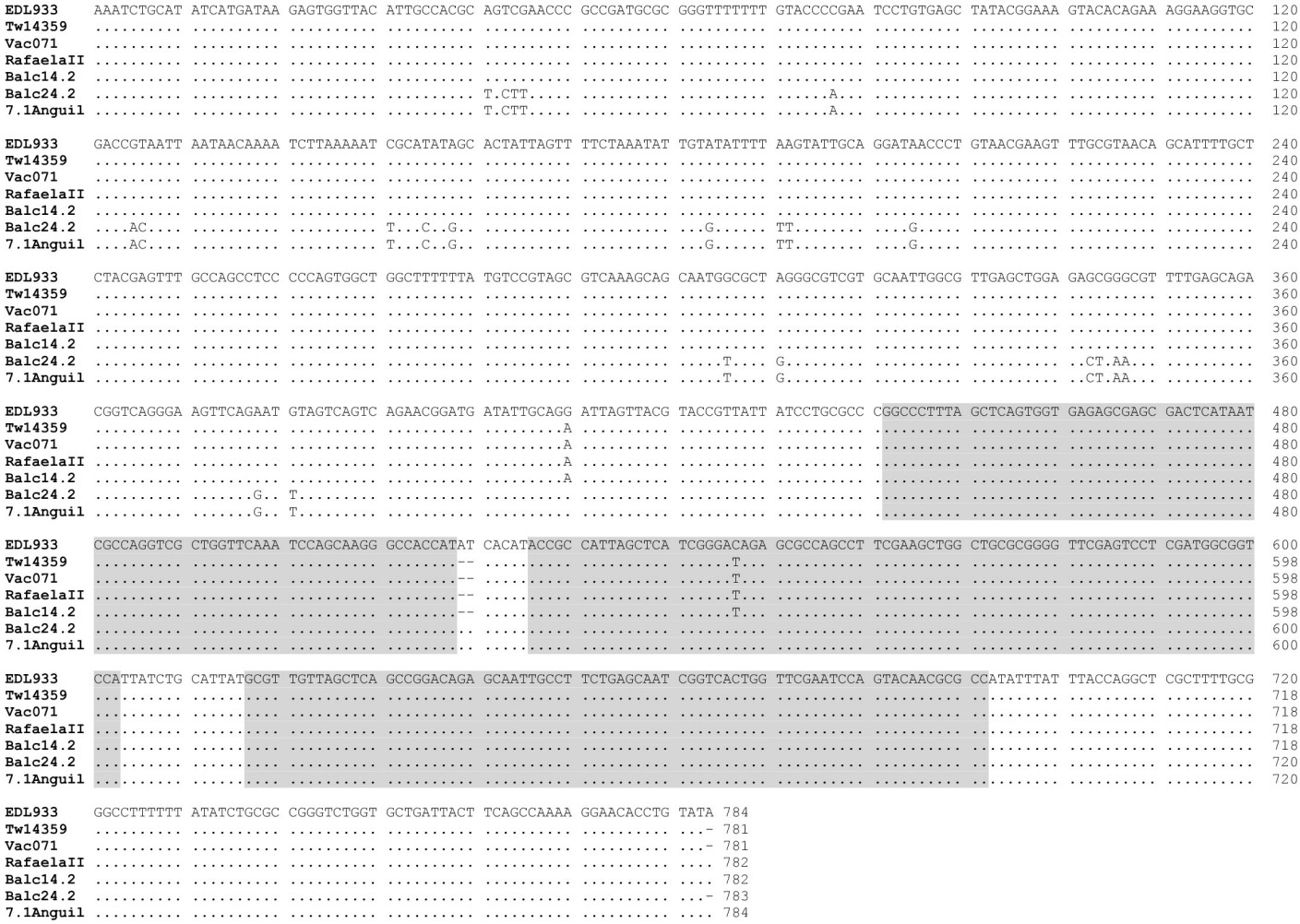

**Fig 6.** (A). amino acid substitutions (shaded in grey) in the Q antiterminator gene, (B) SNPs in *Q-stx2A* intergenic region. EDL933 sequence is used as reference. Dots represent nucleotide that are identical to the reference. SNPs are represented by the substituting nucleotide. A tRNA present in the intergenic region is shaded in grey.

**Identification and variation in virulence genes.** The LEE genes *eae*, *espA*, *espB*, *espF* and *iha* adhesin were 100% conserved among the strains. Notably, *iha* was absent from 438–99 due to a deletion event in that strain.

**Table 4. According to Virulence Finder alleles are named for its accession number[a].**

| Gene | TW14359 | RafaelaII | Balc 14.2 | Vac07.1 | Balc 24.2 | 9.1Anguil | 146N4 | 438–99 | 7.1Anguil | EDL933 |
|------|---------|-----------|-----------|---------|-----------|-----------|-------|---------|-----------|--------|
| *tir* | AE005174 | AE005174 | AE005174 | AE005174 | CP001846 | AE005174 | AE005174 | AE005174 | CP001846 | AE005174 |
| *Eae* | AF071034 | AF071034 | AF071034 | AF071034 | AF071034 | AF071034 | AF071034 | AF071034 | AF071034 | AF071034 |
| *espA* | AE005174 | AE005174 | AE005174 | AE005174 | AE005174 | AE005174 | AE005174 | AE005174 | AE005174 | AE005174 |
| *espB* | AE005174 | AE005174 | AE005174 | AE005174 | AE005174 | AE005174 | AE005174 | AE005174 | AE005174 | AE005174 |
| *espF* | AE005174 | AE005174 | AE005174 | AE005174 | AE005174 | AE005174 | AE005174 | AE005174 | AE005174 | AE005174 |
| *Lha* | AE005174 | AE005174 | AE005174 | AE005174 | AE005174 | AE005174 | AE005174 | Gene deleted | AE005174 | AE005174 |
| *tccP* | CP001368 | CP001368 | CP001368 | CP001368 | AB253545 | CP001368 | CP001368 | AB253537 | AB253545 | CP001368 |
| *espJ* | AE005174 | AE005174 | AE005174 | AE005174 | AE005174 | AE005174 | AE005174 | CP001368 | AE005174 | BA000007 |
| *nleA* | CP001164 | AE005174 | CP001164 | AE005174 | AE005174 | CP001164 | AE005174 | AE005174 | AE005174 | AE005174 |
| *nleB1* | AE005174 | AE005174 | AE005174 | AE005174 | AE005174 | AE005174 | AE005174 | AE005174 | AE005174 | AE005174 |
| *nleB2* | AE005174 | AE005174 | AE005174 | AE005174 | AE005174 | AE005174 | AE005174 | AE005174 | AE005174 | AE005174 |
| *nleC1* | AE005174 | AE005174 | AE005174 | AE005174 | AE005174 | AE005174 | AE005174 | AE005174 | AE005174 | AE005174 |
| *gadA* | BA000007 | BA000007 | BA000007 | BA000007 | BA000007 | BA000007 | BA000007 | BA000007 | BA000007 | BA000007 |
| *gadB* | BA000007 | BA000007 | BA000007 | BA000007 | BA000007 | BA000007 | BA000007 | BA000007 | BA000007 | BA000007 |

[a]accession number corresponds to those informed by virulence finder, rows in yellow mean that all the strains have 100% nucleotide identity, rows in green and red indicate two non- synonymous alleles, whereas rows in green and different grey tone refer to three alleles, being two of them (those in grey) synonymous.

Our analysis detected two *tir* alleles of a previously described T/A SNP at base 255 [64], seven of the Argentinian strains had the T allele while two strains had the A allele (**Table** 4). The effector *tccp1* had three alleles with six strains forming one allele while the other alleles had two and one strain, respectively. The allele (AB253545) with two strains encoded for a protein that is 47-amino acid longer in relation to that encoded for the other two alleles which are synonymous (**Table** 4). Effector *espJ* had three alleles: The most common allele was found in eight of the ten strains and contained a non-sense mutation that created a protein 39 aa shorter at the N-terminal end. The other alleles contained a synonymous mutation (**Table** 4). The effector *nleA* had two synonymous mutations. The *nleB1, nleB2, nleC1 gadA* and *gadB* gene sequences were 100% conserved among strains (**Table** 4). In all strains, *tccp2* was a pseudogene encoding for a non-functional protein.

**LEE genes polymorphisms.** LEE encodes for a T3SS, which is essential in EHEC virulence. The search for polymorphisms of LEE is relevant because we have previously observed important variations in T3SS activity in the same set of strains [20]. For this analysis, we aligned all nucleotide sequences of LEE from all the strains studied here in search of SNPs and In/Dels. Polymorphisms of LEE genes described in the Virulence Finder section will not be described to avoid redundancy.

In all strains, LEE was located in the same genomic position, between rorf1_2 and an IS66 family transposase gene located after the last gene of LEE, *espF*. The LEE sequence was highly conserved among all isolates. Five strains (RafaelaII, Vac07.1, Balcarce24.2, 7.1Anguil and 146N4) presented a substitution at nucleotide (nt) 8797 of LEE, which corresponded to the intergenic region comprised between genes *etgA-grlR* and where the *etgA* and *grlR* promoters should be located. The strain 7.1Anguil strain showed a unique polymorphism in *espA*: a non-synonymous mutation (AACxAAA) producing a NxK change at aa 111.

Among different EHEC O157 strains, LEE locus is much more conserved than prophages and even more than other genome regions [17]. The SNP analysis of the LEE region detected a substitution, between *etgA-grlR* genes, in five strains. The intergenic region is 195 bases in length and is the promoter region for both genes as the *grlR* gene is transcribed in the

downstream direction while the *etgA* genes is transcribed in the upstream direction. GrlR is an important regulatory protein that forms a complex with GrlA regulatory protein [65]. This regulatory system forms a bicistronic operon that coordinates the LEE regulation, where GrlA is the positive regulator and GrlR acts negatively. In addition, other studies showed that the GrlR-GrlA system also regulates the expression of flagella and enterohemolysin in EHEC [66–68]. The *etgA* gene encodes for a peptidoglycan lytic enzyme with muramidase activity, an enzyme that contributes with T3SS needle to go through the peptidoglycan layer of the bacteria so it can be ready to penetrate the host epithelial cell [69].

## Conclusions

In conclusion, clade 8 strains showed conserved genomic structure to each other while other clade strains show inversions and in/dels (>50 kb) with respect to clade 8 strains and, in accordance with previous studies, lambdoid prophage varied among strains, i.e. those encoding for stx2a were extremely polymorphic, in opposition to those encoding for stx2c A cluster of genes presents exclusively in clade 8 contains genes encoding mostly for DNA binding proteins. Stx expression is generally increased in clade 8 strains. However, there are important variations of Stx expression among clade 8 strains that may be associated to the gene content of *stx2a*-prophages and nucleotide substitutions in the promoter region on of *stx2* operon.

## Supporting information

**S1 Fig. Long PCR analysis of 7.1Anguil strain inversion.** A) Evidence and graphic representation primers for combinatorial PCR to evaluate inversion. B) *In silico* prediction of amplicons using assembled genomes. C) Agarose gel of amplified fragments. All primers pairs were tested on 2 isolates from 7.1Anguil and one from RafaelaIIA.
(PDF)

**S2 Fig. Circular genome representation of prophage similarity.** Prophage similarity among genomes of the analyzed strains. Each prophage is represented by a different color on each chromosome. On each circle, genome used as query is showed larger than the subject genomes. Lines starts on query prophages and end on every subject prophage hit showing at least 80% coverage. Size of prophages are represented in scale. For this representation Circos was used.
(PDF)

**S3 Fig. Prophage size vs frequency.** Sizes of prophage families are represented in a scatter graph against the frequency of finding a given prophage family in the 10 genome studied. R2 parameter and the tendency formula were calculated using excel. To determine the relationship between two variables Rho Spearman calculator was used (https://www.socscistatistics.com/tests/spearman/default2.aspx). rho was -0.4341. Shorter prophages were associated with higher clustering ($p<0.01$).
(JPG)

**S1 Table. Identified prophages.** Excel sheets represent all identified prophages. Sheet All: all prophages identified, family or unique, position in the genome, and size data are presented. Sheet family: prophages belonging to families. Sheet of unique prophages: Shared sheet: unique shared types prophages data are presented.
(XLSX)

**S2 Table. Intra strain prophage similarity.** Each sheet represents a different strain and query, subject, Query Coverage Per Subject, Query Coverage Per Unique Subject, Average Identity

and HSPs Number BLAST+ parameters are presented.
(XLSX)

**S3 Table. Main prophage families having > 6 prophages.**
(DOCX)

**S4 Table. Blastn of shared types and unique prophages.** Shared types and Unique prophages described here were searched against nr.
(XLSX)

## Acknowledgments

The authors would like to thank Sandy Fryda-Bradley and the USMARC core sequencing facility for excellent technical assistance. The mention of a trade name, proprietary product, or specific equipment does not constitute a guarantee or warranty by the USDA and does not imply approval to the exclusion of other products that might be suitable. USDA is an equal opportunity provider and employer. AA, ML WMdS MFE and AC are CONICET fellows. MI and NAR holds a CONICET fellowship. We thank Marina Palermo for helping in Stx ELISA determination and Valeria Rocha for her valuable technical help.

## Author Contributions

**Conceptualization:** James L. Bono, David Gally, Angel Cataldi.

**Data curation:** Ariel Amadio, James L. Bono, Mariano Larzábal, Wanderson Marques da Silva, María Florencia Eberhardt, Nahuel A. Riviere, Angel Cataldi.

**Formal analysis:** Matías Irazoqui, María Florencia Eberhardt, Shannon D. Manning, Angel Cataldi.

**Funding acquisition:** Ariel Amadio, James L. Bono, David Gally, Angel Cataldi.

**Investigation:** Ariel Amadio, James L. Bono, Matías Irazoqui, Mariano Larzábal, Wanderson Marques da Silva, María Florencia Eberhardt, Nahuel A. Riviere, Shannon D. Manning, Angel Cataldi.

**Methodology:** Ariel Amadio, James L. Bono, Angel Cataldi.

**Project administration:** Ariel Amadio, Angel Cataldi.

**Resources:** Ariel Amadio, James L. Bono, Mariano Larzábal, Angel Cataldi.

**Software:** Ariel Amadio, James L. Bono.

**Supervision:** Angel Cataldi.

**Validation:** Ariel Amadio.

**Visualization:** Ariel Amadio, James L. Bono, Mariano Larzábal, Wanderson Marques da Silva, Nahuel A. Riviere.

**Writing – original draft:** Ariel Amadio, Angel Cataldi.

**Writing – review & editing:** Ariel Amadio, James L. Bono, Mariano Larzábal, Wanderson Marques da Silva, Nahuel A. Riviere, David Gally, Shannon D. Manning, Angel Cataldi.

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
