## [Decision Letter · Decision Letter 0]

2 Jul 2021

PONE-D-21-18023

Genomic analysis of shiga toxin-containing Escherichia coli O157:H7 isolated from Argentinean cattle

PLOS ONE

Dear Dr. Cataldi,

Thank you for submitting your manuscript to PLOS ONE. After careful consideration, we feel that it has merit but does not fully meet PLOS ONE’s publication criteria as it currently stands. Therefore, we invite you to submit a revised version of the manuscript that addresses the points raised during the review process by both reviewers.

Quality of the figures was borderline when sent out for initial review and I would appreciate if you could include higher quality figures and label them accordingly. Please also address the concerns raised by reviewer 2 in regards to data/software availability. Considering the limited number of isolates analyzed conclusions should not be overstated. As suggested by reviewer 2 "Results/Discussion" could be shortened to avoid redundancies without hurting the overall information content of this manuscript.

We look forward to receiving your revised manuscript.

Kind regards,

Mark Eppinger

Academic Editor

PLOS ONE

Journal Requirements:

5. Please upload a new copy of Figures 1 to 6 as the detail is not clear. Please follow the link for more information: https://blogs.plos.org/plos/2019/06/looking-good-tips-for-creating-your-plos-figures-graphics/" https://blogs.plos.org/plos/2019/06/looking-good-tips-for-creating-your-plos-figures-graphics/

Reviewers' comments:

Reviewer's Responses to Questions

**Comments to the Author**

1. Is the manuscript technically sound, and do the data support the conclusions?

Reviewer #1: Yes

Reviewer #2: Yes

2. Has the statistical analysis been performed appropriately and rigorously? 

Reviewer #1: Yes

Reviewer #2: N/A

3. Have the authors made all data underlying the findings in their manuscript fully available?

Reviewer #1: Yes

Reviewer #2: No

4. Is the manuscript presented in an intelligible fashion and written in standard English?

Reviewer #1: Yes

Reviewer #2: Yes

5. Review Comments to the Author

Reviewer #1: This study reports a comparative genomic analysis of eight EHEC strains isolated from cattle in Argentina, that were previously classified into clades based on SNPs. These strains were sequenced using PacBio technology and used adequate bioinformatic software. They made comparative analysis founding virulence genes related to LEE genomic island, toxin effectors and stress resistance. Among novelty information, the authors found prophage distribution between strains which is relatively different to previous reports and were capable to group into families. Its findings are very interesting and give new insights into diversity of this important pathotype.

I only have a few observations and suggestions to make your manuscript better understood:

1. “Pathogenicity island” is a no longer used term, is preferred “genomic Island” even if it harbors virulence genes, my suggestion is to change the terms.

2. When you mentioned “geneious” you are referring to the software so, you should specify it in order to avoid confusion particularly to novel readers into bioinformatics

3. Shiga-toxin detection and cytotoxicity in Vero cells is a well performed assay and according to previous bibliography and provides important information but it seems to be not related with the title of the manuscript, later in text, the importance of this assay arises so I suggest to justify it when you introduce the results of the experiment

4. All the figures are blurry even if you download it as .tif format, I´m not sure if they are formatted in the correct specification. It makes difficult to follow the text referring to the figure.

5. Lines 229-230. You are talking about cytotoxicity referring to Figure 1, but it is an Artemis map not a toxicity graph

6. Lines 370-375. You are indicating panel (A) and (B) but they are not indicated in the figure, please correct it

Reviewer #2: This manuscript presents a genomic comparison of eight E. coli O157:H7 strains, isolated from the same region of Argentina. Comparative genomic tools were applied to screen for genomic features well-described by this field, including clade typing, prophage analysis, stx2 genotyping, and virulence gene identification. The authors identify characteristics reported before in similar studies, including diverse stx2a prophage but less diversity in stx2c prophage, polymorphisms in stx2-associated genes, and genomic rearrangements. While most of their findings were expected, the experiments were performed properly and for the most part (see below), their interpretations were supported by the data.

Major comments;

1. With hundreds of O157:H7 sequenced genomes available, this seems like a missed opportunity not to perform a larger comparison with other clade 8's. This would add much more value to the paper.

2. In the absence of a deeper genomic analysis, the phrase “sheds light about the genetic basis of hypervirulence” must be removed. This study does not report any experiments that provide new insights into in vivo virulence potential.

3. It appears a decent amount of prophage analysis was done with a “custom Perl script”. Without access to the script, the reader cannot repeat any this portion of the study. This should be made available on GitHub or similar site.

4. What statistical analysis was done to make the “larger prophages” conclusion in lines 261-262?

5. Line 363: Did you test whether this strain makes Shiga toxin using your ELISA? If not, explain in the paper why.

6. The Discussion is a reiteration of the Results. These two sections should be combined.

7. Many of the Figures, even when downloaded via the link provided, are really blurry.

Minor comments:

1. Lines 34-35: sentence fragment

2. Line 66: stx2h and 2i have been reported as well

3. Line 224: temper this language, as there may be additional prophage that weren’t identified by Phaster.

6. PLOS authors have the option to publish the peer review history of their article (what does this mean?). If published, this will include your full peer review and any attached files.

Reviewer #1: **Yes: **Edwin Barrios-Villa

Reviewer #2: No

---

## [Author Response · Author response to Decision Letter 0]

24 Aug 2021

Mark Eppinger

Academic Editor

PLOS ONE

21 August 2021

Ref: PONE-D-21-18023

Genomic analysis of shiga toxin-containing Escherichia coli O157:H7 isolated from Argentinean cattle

Please find attached a revised version of our manuscript: “Genomic analysis of shiga toxin-containing Escherichia coli O157:H7 isolated from Argentinean cattle”. We thank the reviewers for their comments and suggestions and have modified the manuscript accordingly. Please find below a point-by-point response to each of the concerns raised by the reviewers. 

We feel that these changes have significantly enhanced our manuscript, and hope that you will now find it acceptable for publication in Plos One. 

Yours sincerely

Dr. Angel Cataldi

Responses to the academic editor:

Comment/Question (C/Q): Quality of the figures was borderline when sent out for initial review and I would appreciate if you could include higher quality figures and label them accordingly.

Answer (A): We apologize for the low quality of the figures when the pdf to submit was ready. Now we have uploaded high definition tiff figures. 

C/Q: Please also address the concerns raised by reviewer 2 in regards to data/software availability. 

A: all genome data is now available. 

C/Q Considering the limited number of isolates analyzed conclusions should not be overstated. As suggested by reviewer 2 "Results/Discussion" could be shortened to avoid redundancies without hurting the overall information content of this manuscript.

A: As suggested Results and Discussion were combined in a single section. This combination removed much redundancy. Part of the text from the discussion of the original manuscript submitted is distributed in lines 204-207, 233-236, 240-249, 312-317, 320-327, 356-360, 418-432, 444-447, 459-466, and 501-512 of the revised clean manuscript.

Journal Requirements:

A; formatted as requested

A: done as suggested

A: The data is already deposited in GenBank under the BioProject that we’ve provided as private. We now added the accession number for each genome in the manuscript and they will be immediately released as a publication appears. Therefore, we fully comply with PLOS ONE and NCBI policy for Data Availability. 

In the revised version accession number of the genomes (the table below) was placed below Acknowledgements section. 

Strain BioProject BioSample Accession Study SRA Accession

Balcarce_14.2 PRJNA280853 SAMN17295281 CP076243-CP076244 SRP057321 SRR14419381,SRR14419382

Vac07.1 PRJNA280853 SAMN17295280 CP076241-CP076242 SRP057321 SRR14419379,SRR14419380

146N4 PRJNA280853 SAMN17295279 CP076237-CP076240 SRP057321 SRR14419377,SRR14419378

9.1_Anguil PRJNA280853 SAMN17295278 CP076235-CP076236 SRP057321 SRR14419375,SRR14419376

Balcarce_24.2 PRJNA280853 SAMN17295277 CP076245-CP076247 SRP057321 SRR14419387,SRR14419388

7.1_Anguil PRJNA280853 SAMN03470769 CP076232-CP076234 SRP057321 SRR14419385,SRR14419386

Rafaela_II PRJNA280853 SAMN03470766 CP076230-CP076231 SRP057321 SRR14419383,SRR14419384

438/99 PRJNA280853 SAMN17295282 JAHCTZ000000000 SRP057321 SRR14419389,SRR14419390

4. We note that you have included the phrase “data not shown” in your manuscript. 

A: we replaced the first data no shown (line 274 of the revised clean version) by S Table 3, and the second (line 470) was deleted, as text is itself self-explanatory. 

5. Please upload a new copy of Figures 1 to 6 as the detail is not clear. Please follow the link for more information: https://blogs.plos.org/plos/2019/06/looking-good-tips-for-creating-your-plos-figures-graphics/" https://blogs.plos.org/plos/2019/06/looking-good-tips-for-creating-your-plos-figures-graphics/

A; new more defined figures were uploaded

Reviewers' comments and Reviewer's Responses to Questions

 Reviewer #1: This study reports a comparative genomic analysis of eight EHEC strains isolated from cattle in Argentina, that were previously classified into clades based on SNPs. These strains were sequenced using PacBio technology and used adequate bioinformatic software. They made comparative analysis founding virulence genes related to LEE genomic island, toxin effectors and stress resistance. Among novelty information, the authors found prophage distribution between strains which is relatively different to previous reports and were capable to group into families. Its findings are very interesting and give new insights into diversity of this important pathotype.

I only have a few observations and suggestions to make your manuscript better understood:

1. “Pathogenicity island” is a no longer used term, is preferred “genomic Island” even if it harbors virulence genes, my suggestion is to change the terms.

A: corrected as requested

2. When you mentioned “geneious” you are referring to the software so, you should specify it in order to avoid confusion particularly to novel readers into bioinformatics

A: Yes, we are referring to software Geneious. We believed it was clear that was a product because we added the company in brackets, but we added the word “software” to clarify.

3. Shiga-toxin detection and cytotoxicity in Vero cells is a well performed assay and according to previous bibliography and provides important information but it seems to be not related with the title of the manuscript, later in text, the importance of this assay arises so I suggest to justify it when you introduce the results of the experiment

A: We acknowledge the comment of the reviewer and we add this sentence in lines 401-403 (of the revised clean version) “Stx toxin production was analyzed in strains carrying stx2a operon (Fig 5) in order to associate gene content of the encoding prophages with Stx production” 

4. All the figures are blurry even if you download it as .tiff format, I´m not sure if they are formatted in the correct specification. It makes difficult to follow the text referring to the figure.

A: The reviewer is right. More defined figures were uploaded for the revised version

5. Lines 229-230. You are talking about cytotoxicity referring to Figure 1, but it is an Artemis map not a toxicity graph

W: we apologize for the mistake. We deleted the sentence about Verocitotoxicity assays and as this information (line 229-230) is redundant with that in Lines 363-368. We prefer to leave it as it is in 363-368 (lines 405-410 in the revised version)

6. Lines 370-375. You are indicating panel (A) and (B) but they are not indicated in the figure, please correct it

A: corrected as indicated 

Reviewer #2: This manuscript presents a genomic comparison of eight E. coli O157:H7 strains, isolated from the same region of Argentina. Comparative genomic tools were applied to screen for genomic features well-described by this field, including clade typing, prophage analysis, stx2 genotyping, and virulence gene identification. The authors identify characteristics reported before in similar studies, including diverse stx2a prophage but less diversity in stx2c prophage, polymorphisms in stx2-associated genes, and genomic rearrangements. While most of their findings were expected, the experiments were performed properly and for the most part (see below), their interpretations were supported by the data.

Major comments;

1. With hundreds of O157:H7 sequenced genomes available, this seems like a missed opportunity not to perform a larger comparison with other clade 8's. This would add much more value to the paper.

A: We understand that a larger dataset of E. coli genomes is available, however, most of the comparisons made in the present study included some phenotypic information of the analyzed strains or manual comparisons (like alignments) of the data that couldn’t be performed in a dataset of thousands of genomes. We chose to sequence the genomes to completion and use complete closed STEC O157:H7 genomes from NCBI in our analysis because we wanted to compare all aspects of genome diversity including chromosome architecture, complete phage content, plasmids and other mobile genome elements to identify any features that might associate with observed phenotype difference between the Argentinean strains and also those used in the comparison. Such detailed resolution is not available with draft genome sequences, which are the majority on public databases. However, we believe that the present study could be a start point to think how to upgrade comparisons made to include all deposited E. coli genomes and we intend to do this in future studies.

2. In the absence of a deeper genomic analysis, the phrase “sheds light about the genetic basis of hypervirulence” must be removed. This study does not report any experiments that provide new insights into in vivo virulence potential.

A: the sentence was changed to “This genomic analysis may contribute to the understanding of the genetic basis of the hypervirulence of EHEC O157:H7 strains circulating in Argentine cattle”.

3. It appears a decent amount of prophage analysis was done with a “custom Perl script”. Without access to the script, the reader cannot repeat any this portion of the study. This should be made available on GitHub or similar site.

A: The Perl script was only used to parse BLAST comparisons files. But as the reviewer suggested, was uploaded to GitHub repository to facilitate reproduction. It is now available at https://github.com/arielamadio/Ecoli_parsing_data and this was also added to the manuscript.

4. What statistical analysis was done to make the “larger prophages” conclusion in lines 261-262?

A: the text about larger prophages associated to unique prophages was moved the end of the Prophage diversity section in lines. We added a text: A tendency was observed that shorter prophages are likely to be associated with higher clustering (S Fig 3), as previously described by Shaaban et al [43]. Conversely, larger prophages (>80 kb) were overrepresented among the unique prophages (line 307-309 in the revised clean version). 

The association of prophage size with clustering was determined by Spearman Rho test that the association was statistically significant. 

5. Line 363: Did you test whether this strain makes Shiga toxin using your ELISA? If not, explain in the paper why.

A: Yes, in ELISA E. coli O157:H7 9.1 Anguil appears as a very low producer while E. coli O157:H7 RafII is a strong producer as shown in our previous paper (Amigo et al, PLoS One. 2015;10. doi:10.1371/journal.pone.0127710, Fig S1).

6. The Discussion is a reiteration of the Results. These two sections should be combined.

W: As indicated by thee reviewer the two sections were combined. 

7. Many of the Figures, even when downloaded via the link provided, are really blurry.

W. new more defined figures were uploaded

Minor comments:

1. Lines 34-35: sentence fragment

A: corrected as suggested. Sentence was fragmented to: Clade 8 strains that were classified as hypervirulent. Most of the strains of this clade have a Shiga toxin stx2a-stx2c genotype

2. Line 66: stx2h and 2i have been reported as well

A: corrected as suggested to stx2 (stx2a, stx2b, stx2c, stx2d, stx2e, stx2f, stx2g, stx2h and stx2i) 

3. Line 224: temper this language, as there may be additional prophage that weren’t identified by Phaster.

A: Corrected to The number of inserted prophages detected by Phaster, varied from 13 to 16 per genome

---

## [Decision Letter · Decision Letter 1]

17 Sep 2021

PONE-D-21-18023R1

Genomic analysis of shiga toxin-containing Escherichia coli O157:H7 isolated from Argentinean cattle

PLOS ONE

Dear Dr. Cataldi,

Please address the comment raised by one of the reviewers in regards to the discussion of stx2 suballeles and references.

"The authors added "stx2h" and "stx2i" as previously suggested, but use an incorrect 2012 reference (these variants were first reported around 2018-2019).  Please see doi:10.1038/s41598-018-25233-x and doi: https:doi.org/10.4315/ 0362-028X.JFP-18-291."

Also, a more recent article reports on stx2k (Yang, X. et (2020). Escherichia coli strains producing a novel Shiga toxin 2 subtype circulate in China. Int J Med Microbiol 310, 151377.).

We look forward to receiving your revised manuscript.

Kind regards,

Mark Eppinger

Academic Editor

PLOS ONE

Journal Requirements:

Additional Editor Comments (if provided):

Reviewers' comments:

Reviewer's Responses to Questions

**Comments to the Author**

1. If the authors have adequately addressed your comments raised in a previous round of review and you feel that this manuscript is now acceptable for publication, you may indicate that here to bypass the “Comments to the Author” section, enter your conflict of interest statement in the “Confidential to Editor” section, and submit your "Accept" recommendation.

Reviewer #1: All comments have been addressed

Reviewer #2: (No Response)

2. Is the manuscript technically sound, and do the data support the conclusions?

Reviewer #1: Yes

Reviewer #2: Yes

3. Has the statistical analysis been performed appropriately and rigorously? 

Reviewer #1: Yes

Reviewer #2: N/A

4. Have the authors made all data underlying the findings in their manuscript fully available?

Reviewer #1: Yes

Reviewer #2: Yes

5. Is the manuscript presented in an intelligible fashion and written in standard English?

Reviewer #1: Yes

Reviewer #2: Yes

6. Review Comments to the Author

Reviewer #1: As mentioned in past review, this study worth to be communicated. All recommendations have been addressed

Reviewer #2: The authors added "stx2h" and "stx2i" as previously suggested, but use an incorrect 2012 reference (these variants were first reported around 2018-2019). Please see doi:10.1038/s41598-018-25233-x and doi: https:doi.org/10.4315/ 0362-028X.JFP-18-291.

7. PLOS authors have the option to publish the peer review history of their article (what does this mean?). If published, this will include your full peer review and any attached files.

Reviewer #1: No

Reviewer #2: No

---

## [Author Response · Author response to Decision Letter 1]

30 Sep 2021

Mark Eppinger

Academic Editor

PLOS ONE

30 September 2021

Ref: PONE-D-21-18023 R1

Genomic analysis of shiga toxin-containing Escherichia coli O157:H7 isolated from Argentinean cattle

Please find attached a revised (R2) version of our manuscript: “Genomic analysis of shiga toxin-containing Escherichia coli O157:H7 isolated from Argentinean cattle”. We thank the reviewers for their comments and suggestions and have modified the manuscript accordingly. Please find below a point-by-point response to each of the concerns raised by the reviewers. 

We feel that these changes have significantly enhanced our manuscript, and hope that you will now find it acceptable for publication in Plos One. 

Yours sincerely

Dr. Angel Cataldi

Responses to the academic editor:

Comment/Question (C/Q):"The authors added "stx2h" and "stx2i" as previously suggested, but use an incorrect 2012 reference (these variants were first reported around 2018-2019). Please see doi:10.1038/s41598-018-25233-x and doi: https:doi.org/10.4315/ 0362-028X.JFP-18-291."

Answer (A): the two reference were added in line 67 as [8] and [9].

C7Q: Also, a more recent article reports on stx2k (Yang, X. et (2020). Escherichia coli strains producing a novel Shiga toxin 2 subtype circulate in China. Int J Med Microbiol 310, 151377.).

A: The reference was added as [10] in line 67

. Review Comments to the Author

C7Q: Reviewer #1: As mentioned in past review, this study worth to be communicated. All recommendations have been addressed

A: questions of Reviewer 2 have been addressed under Responses to the academic editor:

---

## [Editor Report · Decision Letter 2]

5 Oct 2021

Genomic analysis of shiga toxin-containing Escherichia coli O157:H7 isolated from Argentinean cattle

PONE-D-21-18023R2

Dear Dr. Cataldi,

We’re pleased to inform you that your manuscript has been judged scientifically suitable for publication and will be formally accepted for publication once it meets all outstanding technical requirements.

Kind regards,

Mark Eppinger

Academic Editor

PLOS ONE
---

## [Editor Report · Acceptance letter]

19 Oct 2021

PONE-D-21-18023R2 

Genomic analysis of shiga toxin-containing *Escherichia coli* O157:H7 isolated from Argentinean cattle 

Dear Dr. Cataldi:

I'm pleased to inform you that your manuscript has been deemed suitable for publication in PLOS ONE. Congratulations! Your manuscript is now with our production department. 

Kind regards, 

on behalf of

Dr. Mark Eppinger 

Academic Editor

PLOS ONE